# The Relationship between Physical Fitness and Cognitive Functions in Older People: A Systematic Review

**Maria Antonieta Tinôco** [1], **Marcelo de Maio Nascimento** [2], **Adilson Marques** [3,4,*], **Élvio Rúbio Gouveia** [5,6,*], **Salvador Miguel** [5], **Francisco Santos** [5] **and Andreas Ihle** [7,8,9]

1   Coordination of Physical Education and Sport, Federal Institute of Science and Technology Education of Amazonas, Manaus 69020-120, Brazil; nittin@me.com
2   Department of Physical Education, Federal University of Vale do São Francisco, Petrolina 56304-917, Brazil; marcelo.nascimento@univasf.edu.br
3   Centre for the Study of Human Performance (CIPER), Faculty of Human Kinetics, University of Lisbon, 1495-751 Lisbon, Portugal
4   Instituto de Saúde Ambiental (ISAMB), Faculty of Medicine, University of Lisbon, 1649-020 Lisbon, Portugal
5   Department of Physical Education and Sport, University of Madeira, 9020-105 Funchal, Portugal; salvador192001@hotmail.com (S.M.); francisco191santos@gmail.com (F.S.)
6   Laboratory of Robotics and Engineering Systems (LARSYS), Interactive Technologies Institute, 9020-105 Funchal, Portugal
7   Department of Psychology, University of Geneva, 1205 Geneva, Switzerland; andreas.ihle@unige.ch
8   Center for the Interdisciplinary Study of Gerontology and Vulnerability, University of Geneva, 1205 Geneva, Switzerland
9   Swiss National Centre of Competence in Research LIVES—Overcoming Vulnerability: Life Course Perspectives, 1015 Lausanne, Switzerland
*   Correspondence: amarques@fmh.ulisboa.pt (A.M.); erubiog@staff.uma.pt (É.R.G.)

**Abstract:** The ageing process is associated with vulnerabilities, such as cognitive decline. Physical activity and exercise are key for preserving cognitive health in older age. This systematic review aims to analyse the effects of physical fitness programs on healthy older adults' cognitive functions. An electronic search was performed in the PubMed, Web of Science, and Scopus databases. It included observational and experimental studies published between February 2017 and March 2023. Of the 1922 studies identified, 38 met the inclusion criteria. The findings show the positive effects of physical training on cognitive function in older adults. The most examined cognitive domains were executive function, memory function, and global cognition. Aerobic training prevailed, followed by resistance strength training and exergames. There was high variability in the characteristics of the protocols. The average length of interventions was 3–6 months; the frequency varied in the range of 1–4-times a week and 30–90 min sessions. The findings of this systematic review emphasise that physical fitness programs positively improve the specific domains of cognitive function in healthy older adults. These results can contribute to planning future interventions to improve the mental health of the older population and strengthen the development of policies for healthy ageing.

**Keywords:** cognitive function; physical fitness; older adults; active ageing

## 1. Introduction

The ageing process is associated with vulnerabilities, such as cognitive decline [1,2] and increased non-communicable diseases [3]. The association between cognitive decline and comorbidities increases the chances of older individuals experiencing barriers to adapting to the environment and an increased risk of death [4,5]. Review studies and meta-analyses have shown that, at an advanced age, regular physical activity (PA) is a strategy capable of mitigating changes in cognitive function (CF) [6,7]. PA can trigger positive cognitive stress reflexes [8], which in turn generate brain changes (neural plasticity), resulting in a better activation of neurons and facilitating new demands and behavioural adaptations [9].

PA's role in the CFs of older individuals underlies the improvement in their physical fitness (PF). Regular PA practice can substantially increase PF levels or one of its components, namely, cardiorespiratory fitness, muscular strength, endurance, balance, flexibility, speed, agility, coordination, and body composition [10]. In turn, an increase in cardiorespiratory fitness can stimulate the processes underlying neurogenesis in older adults [11], promoting neuroplasticity in the hippocampus [12], which consequently benefits executive functions (EFs) [13]. Thus, when older adults practice regular PA, spatial learning induced by activities improves their memory performance [14]. Furthermore, in later life, which is considered as the period that generally begins at retirement age, around 60 or 70 years old, and extends to the end of people's lives, maintaining sufficient levels of muscle strength is crucial for performing daily activities [15]. Therefore, it is advisable for older adults to engage in weekly resistance training sessions, potentially leading to improvements in their cognitive function [16]. However, the combination of resistance training and an aerobic intervention creates additional benefits for CF compared to aerobic training alone [17]. A current study has shown that this conclusion requires investigations that compare the effects of low- and high-intensity types of exercises on the neuroplasticity of the older adult population [18].

With the improvement in PF components through PA, neurogenesis is triggered at the structural level, resulting from cell proliferation and dendritic branching [14]. Another CF potentiating factor that PA can release is the neurotrophic action of the brain-derived neurotrophic factor (BDNF), a mechanism known to act in the structural alteration of the central nervous system (CNS) [19]. The BDNF can benefit peripheral systems, favouring the health of older adults, as it reduces food intake and increases the glucose oxidation rate and insulin sensitivity [20].

Over the years, several investigations have been conducted to test, identify, and determine the effects of PA or PF on improving CFs in older adults. Thus, different protocols were tested, such as aerobic exercises [12,21], resistance training [22,23], and multicomponent training that combines both strength training and aerobic exercises with other training modalities [24]. However, it remains inconclusive which CF domains are most responsive to PF programs, which types of training are the most effective in generating neuroplasticity, and what the ideal weekly frequency and total duration of a program should be [6,25]. Consequently, it is necessary to summarise the different guidelines on the type, frequency, and intensity of PF that should be prescribed to benefit older adults' CFs [16]. Therefore, we conduct a systematic review of both observational and experimental studies to examine the influence of physical fitness programs on healthy older adults' cognitive functions. Our specific objectives are as follows: (i) to ascertain the most commonly employed types of PF training and their effects on different CF domains; (ii) to identify the specific cognitive functions assessed, along with the instruments or tasks used for the evaluation; and (iii) to provide a comprehensive overview of the training or tasks performed during the interventions, including frequency, duration, and session duration.

## 2. Methods

### 2.1. Study Design

The present study was conducted according to the Preferred Reporting Items for Systematic Reviews and Meta-Analyses (PRISMA) 2020 guidelines [26]. Figure 1 illustrates the PRISMA checklist. This review was registered in Prospero, whose registration number is CRD42022314794.

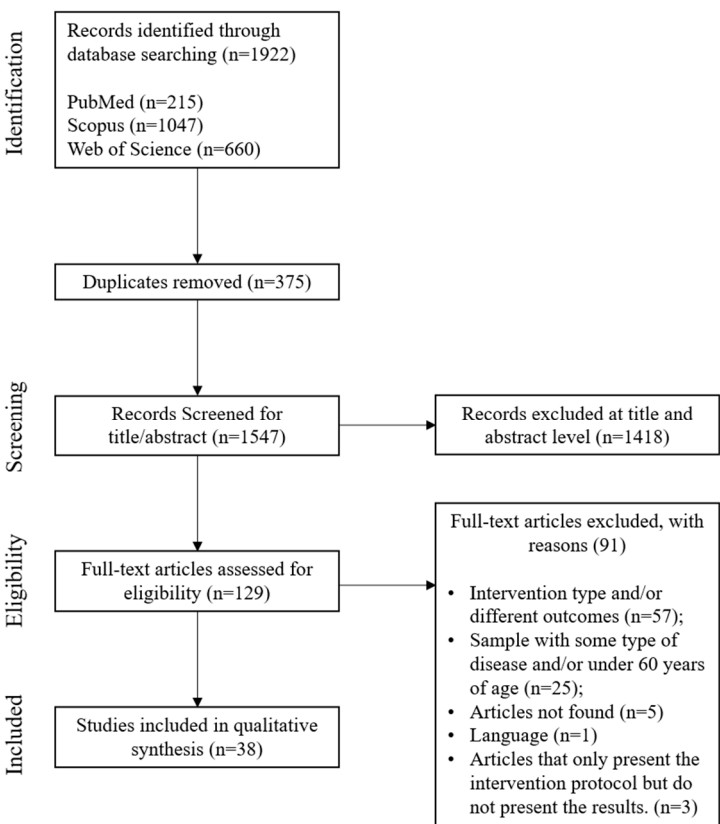

**Figure 1.** Flowchart of the study selection.

*2.2. Search Strategy*

The lead author conducted a comprehensive search across three electronic databases (PubMed, Web of Science, and Scopus) in March 2023. We limited our inclusion criteria to articles exploring the impact of PF programs on CFs in elderly individuals, specifically focusing on peer-reviewed scientific journals published between February 2017 and March 2023. The choice to consider articles from 2017 onwards was influenced by the existence of a similar review published in that year [6].

The manuscripts included in the present review met the following criteria: (1) population—healthy older adults (≥60 years old) without any associated disease; (2) intervention—PF programs; (3) comparator—studies with and without comparison/control groups; (4) outcome—cognitive function; and (5) studies—observational and experimental. The exclusion criteria were: (1) studies that presented a sample with an associated disease because there are certain conditions, medications, or other comorbidities that could affect cognition, and this could influence the results regarding the impact of physical fitness programs on cognitive function; (2) articles published before 2017; and (3) articles published in languages other than English. The following terms were searched for in the title and abstract: ("Cognitive function" OR "Cognitive dysfunction" OR "Cognitive behavio*" OR "Cognitive decline" OR "Cognitive domains" OR Dement*) AND ("Physical activity" OR Exercise OR Sport OR Fitness OR Functional OR Movement) AND (Healthy OR "Active ageing") AND ("Older adults" OR Senior OR Elder* OR "Older people"). The search terms were defined based on the previous systematic review [6] and after agreement among the authors (Table 1).

**Table 1.** Search terms and keywords used in each database.

| Key Search Terms | Related Search Terms |
|---|---|
| Cognitive function | "Cognitive function" OR "Cognitive dysfunction" OR "Cognitive behavio*" OR "Cognitive decline" OR "Cognitive domains" OR Dement * |
| Physical fitness | "Physical activity" OR Exercise OR Sport OR Fitness OR Functional OR Movement |
| Active ageing | Healthy OR "Active ageing" |
| Older adults | "Older adults" OR Senior OR Elder * OR "Older people" |

Note. The asterisk (*) represents truncations.

### 2.3. Screening Strategy and Study Selection

Once the search was completed, all returned studies were combined and exported into a reference manager software (EndNote X20, Thomson Reuters, Philadelphia, PA, USA) for further evaluation. Duplicates were automatically removed and manually checked. Three authors (M.A.T., F.S., and S.M.) independently checked the title and abstract for eligibility. The inclusion and exclusion decisions were determined by consensus among the same authors after they had read all the eligible records.

### 2.4. Data Extraction and Harmonisation

Table 2 contains the primary information of the studies included in this systematic review. The data extraction and harmonisation were performed by the three authors (M.A.T., F.S., and S.M.), who grouped all the relevant information, summarising the sample characteristics (sample number and mean age), the purpose of the study, the setting and country where it was applied, the type of physical training applied, the weekly frequency and duration, the primary outcomes and the instruments used to assess the cognitive domains, and the main results.

**Table 2.** Characteristics and main results of the studies included in the systematic review.

| Authors and Year | Sample Characteristics | Purpose | Setting and Country | Intervention (Duration; Main Characteristics of the Program) | Cognitive Outcome (Measures) | Main Results |
|---|---|---|---|---|---|---|
| [27] (2020) | 64 F (IG = 32, aged mean 65.48 years old; CG = 32, aged mean 64.77 years old). | Evaluate the effect of "Brain Gym" exercises on cognitive function and plasma brain-derived neurotrophic factor in the elderly. | Public health centre in Jakarta, Indonesia. | IG: 12 weeks; aerobic (mostly core movements); 2×/week; 60 min of exercise. CG: exercise not performed. | Orientation, memory, recall, attention, naming objectives, verbal and written commands, writing and coping with a figure (MMSE) | No significant difference was seen in MMSE scores between treatment and control groups. |
| [28] (2018) | 149 F + 72 M (IG1 = 49, aged mean 64.82 years old; IG2 = 49, aged mean 64.51 years old; IG3 = 59, aged mean 65.58 years old; IG4 = 64, aged mean 65.52 years old). | Examined whether baseline brain network modularity predicted cognitive improvements in older adults after an exercise intervention. | Urbana-Champaign community in Illinois, USA. | IG1: 6 months; aerobic (walking); 3×/week; 60 min of exercise. IG2: 6 months; aerobic (walking + supplementation); 3×/week; 60 min of exercise. IG3: 6 months; stretching, strengthening and stability; 3×/week; 60 min of exercise. IG4: 6 months; aerobic (dance); 3×/week; 60 min of exercise. | Vocabulary, perceptual speed, episodic memory, and fluid reasoning (VCAP) Executive function (switching task and spatial working memory). | IG1, IG2, and IG3 showed greater executive function gains compared to the IG4 group. No group effects were observed for improvements in perceptual speed, episodic memory, and vocabulary. |

**Table 2.** *Cont.*

| Authors and Year | Sample Characteristics | Purpose | Setting and Country | Intervention (Duration; Main Characteristics of the Program) | Cognitive Outcome (Measures) | Main Results |
|---|---|---|---|---|---|---|
| [29] (2023) | 79 F + 41 M (IG = 59, aged mean 70.6 years old; CG = 61, aged mean 72 years old). | Examined the cognitive effects of the home-based computerised multidomain intervention StayFitLonger (SFL), combining physical exercise and cognitive training, compared to an active control condition. | Home in Switzerland, Canada, and Belgium. | IG (StayFitLonger—computerised home-based training). 26 weeks; strength, balance, and mobility + cognitive training; at least 3×/week; 30–45 min exercise. CG: the active control intervention had structure, timing, and organisation similar to the IG. The difference was that they only had a limited number of physical exercises and did not include interactive videos, personalisation, chat rooms, psycho-educational content, or a virtual guide. | Global Cognition (ZAVEN) Executive Function (Letter Fluency Test, TMT, VST and TAP) Memory (CVLT and Logical Memory Task) Processing Speed (TMT, DSST and VST). | They found that pre-frail individuals in the IG improved their global cognition and processing speed scores after the intervention, unlike participants in the active control condition and unlike robust participants enrolled in either intervention. |
| [30] (2021) | 23 F + 9 M (IG1 = 17, aged mean 76.5 years old; IG2 * = 15, aged mean 80.5 years old). | Determine the effects of exercise training on cognition and cortical grey matter microstructure in individuals with MCI vs. cognitively healthy older adults. | Community (retirement communities and recreation centres) in Wisconsin, USA. | IG1 12 weeks; aerobic (treadmill walking); 3×/week; 30 min exercise. IG2: not considered *. | Multiple aspects of cognition (Geriatric Depression Scale, DRS-2, RAVLT, COWAT, Semantic Animal Fluency Test, and the Clock Drawing Test). | IG significantly improved in RAVLT (verbal memory) and COWAT (verbal fluency). |
| [31] (2021) | 63 F + 42 M (IG = 52, aged mean 71.8 years old; CG = 53, aged mean 72.7 years old). | Verify whether medium-intensity physical activity in elderly people living in the community effectively improves cognitive performance. | Community in Cagliari, Italy. | IG: 12 weeks; aerobic, anaerobic, strength, and balance exercises; 3×/week; 65 min exercise. CG: same time of exposure, but this group performed activities focusing on the history of local culture and education of wellness. | Attention, memory, verbal fluency, language, visual-spatial skills (ACE-R) | IG showed improvements in the ACE-R and better performances for the memory and visual-space skills subscales of the ACE-R. |
| [32] (2022) | 18 F + 7 M (IG1 = 13, aged mean 60.3 years old; IG2 = 12, aged mean 70.2 years old). | Compared the effects of traditional resistance training and resistance training combined with cognitive tasks on body composition, physical performance, cognitive function, and plasma brain-derived neurotrophic factor levels in older adults. | Community and does not represent the country. | IG1 (resistance training + cognitive tasks); 16 weeks; resistance (major muscle groups using machines, free weights, and body weight); 2×/week; 60 min of exercise + verbal fluency cognitive task. IG2 (resistance training only). Same resistance training as IG1. | Verbal fluency (VFT) Dual task (TUG + cognitive test) Short-term memory (SPMT) executive function, visual attention, and task switching (TMT). | Exclusive improvements in cognitive function were observed only in IG1. The improvements were only in verbal fluency and dual-task tests. Short-term memory and executive function did not present significant improvements. |

**Table 2.** *Cont.*

| Authors and Year | Sample Characteristics | Purpose | Setting and Country | Intervention (Duration; Main Characteristics of the Program) | Cognitive Outcome (Measures) | Main Results |
|---|---|---|---|---|---|---|
| [33] (2020) | 16 F + 12 M (IG1 = 10, aged mean 68.8 years old; IG2 = 9, aged mean 71.2 years old; CG = 9, aged mean 67.7 years old) | Explored the effects of exercise with either a high cognitive load (IG1) or low cognitive load (IG2) on cognitive performance and neuroplasticity in healthy elderly. | Community in Taiwan, China. | IG1: 4 months; aerobic (dance); 3×/week; 50 min exercise. IG2: 4 months; aerobic (walking on treadmill); 3×/week; 50 min exercise. CG: exercise not performed. | Attention, orientation, short-term memory, long-term memory, language, drawing, abstract thinking and judgment, mental manipulation, and animal name fluency (CASI 2.0). | IG1 had a significantly higher score on the CASI test demonstrating that a high-cognitive load, but not exercise with a low-cognitive load, improved the overall cognitive function of healthy, elderly individuals. |
| [34] (2019) | N/S F + N/S M (IG1= 30, aged mean 65.5 years old; IG2 * = 30, aged mean 67.5 years old). | Demonstrated the importance of traditional Greek dances in improving both the cognitive and physical health of the senior citizens. | Greek Association of Alzheimer's Disease and Relative Disorders and Daycare Centres in Thessaloniki, Greece. | IG1 24 weeks; aerobic (traditional Greek dance); 2×/week; 60 min exercise. IG2: not considered *. | Cognitive performance (CDR), attention (TEA), executive function (TMT and FUCAS), memory (ROCF, RAVLT, and RBMT), verbal fluency (VFT). | After the intervention was verified, positive changes occurred for attention, executive function, immediate memory, and delayed recall. |
| [35] (2020) | 41 F + 27 M (IG1 = 21, aged mean 71.3 years old; IG2 = 24, aged mean 69.5 years old; IG3 = 23, aged mean 69.9 years old). | Tested the effects of unstable vs. stable resistance training on executive functions. | Community in Kassel, Germany. | IG1: 10 weeks; resistance (instability free weights); 2×/week; 60 min exercise. IG2: 10 weeks; resistance (stable machine-based exercise); 2×/week; 60 min exercise. CG: 10 weeks; resistance (stable machine-based adductor/abductor training); 2×/week; 60 min exercise. | Neuropsychological (DSST), memory (DMT), selective attention and processing speed (Stroop), visual attention and task switching (TMT). | IG1 (instability resistance training), improved working memory, processing speed, and response inhibition. In contrast, improvements in executive functions for IG2 and IG3 were not verified. |
| [36] (2020) | 31 F + 9 M (IG1 = 12, aged mean 68.08 years old; IG2 = 15, aged mean 67.2 years old; CG = 14, aged mean 67.2 years old). | Compared the effects of dance/movement training to aerobic exercise training on cognition, physical fitness, and health-related quality of life in healthy, inactive, elderly individuals. | Community in Canada. | IG1: 12 weeks; aerobic (dance movement); 3×/week; 60 min exercise. IG2: 12 weeks; aerobic (recumbent bicycle); 3×/week; 60 min exercise. CG: exercise not performed. | Executive functions (dual-task, N-back, and Digit Stroop); global cognition (MoCA) | Executive and non-executive composite scores significantly increased, but no group difference or interaction was verified. There was no time effect, group difference, or interaction for the MoCA. |

**Table 2.** *Cont.*

| Authors and Year | Sample Characteristics | Purpose | Setting and Country | Intervention (Duration; Main Characteristics of the Program) | Cognitive Outcome (Measures) | Main Results |
|---|---|---|---|---|---|---|
| [37] (2021) | 22 F + 9 M (IG = 15, aged mean 67.6 years old; CG = 16, aged mean 69.1 years old). | Investigated the effect of a physical exercise multicomponent training based on exergames on cognitive function in older adults. | Local senior gymnasium in Madeira, Portugal. | IG 12 weeks; functional exercise (group training sessions) + exergames fitness program; 2×/week (1-day functional fitness + 1-day exergames); 45 min exercise. CG: 12 weeks; functional exercise (group training sessions); 2×/week; 45 min exercise. | Prospective memory, verbal short-term memory, long-term memory, working memory, verbal fluency, and inductive reasoning (COGTEL). | Both groups obtained significant short-term memory, long-term memory, and COGTEL total scores. However, at the follow-up session (4 weeks later), only the IG group showed a significant improvement in short-term memory and long-term memory. |
| [38] (2020) | 105 F + 101 M aged mean 65.9 years old. | Investigated the effects of a 6-month aerobic exercise intervention on cognition and cerebrovascular regulation. | Community in Calgary, Alberta, Canada. | 6 months; aerobic (aerobic training); 3×/week; the exercise duration increased from 20 to 40 min as participants progressed in the program. | Processing speed (Symbol Digit Modalities Test), executive functions (Cart Sorting Test and Colour and Word Inference Test), verbal memory (Buschke Selective Reminding Test), figural memory (Medical College of Georgia Complex Figure), fluency (Verbal Fluency Test), attention (Auditory Consonant Trigram Test). | After the intervention, positive changes were seen in the executive functions (processing speed and concept formation), verbal memory, and fluency. The figural memory domain showed a negative change, and no changes were found for complex attention. |
| [39] (2021) | 24 F + 14 M (IG1 = 20, aged mean 69.9 years old; IG2 = 18, aged mean 70.9 years old). | Whether a combination of Bifidobacterium spp. supplementation and moderate resistance training improved cognitive function and other health-related parameters in healthy, elderly subjects. | Public liberal art school in Hyogo, Japan. | IG1: probiotic group: 12 weeks; resistance training (combination of latex band training, squats, and Tai Chi); 90 min. Participants were encouraged to perform daily exercises at home, such as stretching, squatting, or walking, and record the types and durations of each exercise session. IG 2: placebo group: same training as IG1. | Global cognition (MoCA); response accuracy reaction time (Flanker task test). | MoCA scores showed a significant increase in both groups, while the flanker task scores for the probiotic group increased more significantly than those for the placebo group. |

**Table 2.** *Cont.*

| Authors and Year | Sample Characteristics | Purpose | Setting and Country | Intervention (Duration; Main Characteristics of the Program) | Cognitive Outcome (Measures) | Main Results |
|---|---|---|---|---|---|---|
| [40] (2021) | 61 F + 11 M (IG = 41, aged mean 67.39 years old; CG = 31, aged mean 67.87 years old). | Analysed the effects of dual-task multimodal physical exercise at a moderate intensity and cognitive stimulation on cognitive and physical functions on healthy, older adults. | Community in Brazil. | IG1: exercise + cognitive stimulation (dual task), 12 weeks; aerobic, resistance and flexibility training (walking, functional circuits, agility, balance, coordination, dance, and global resistance exercises, such as squats and bench press); 2×/week; 75 min exercise. CG: educational materials on health-related topics. | Cognitive performance (CANTAB), ability to understand and complete tasks (Motor Screening Test), episodic memory (PAL test), visual attention (RVT test). | Intervention positively influenced episodic memory (PAL test) and sustained visual attention (RVP test). |
| [41] (2020) | 53 F + 43 M (IG1 = 24, aged mean 65.45 years old; G2 = 25, aged mean 66.52 years old; IG3 = 22, aged mean 65.53 years old; CC = 25, aged mean 66.34 years old). | Investigated the effects of physical exercise tapping high-level cognitive functions on both cognitive function and fitness in older adults. | Community in Shanghai, China. | IG1: the same training as IG2 + cognitive training. IG2: 12 weeks; aerobic training (treadmill walking); 3×/week; 60 min exercise. IG3: cognitive training—program designed to improve executive function and memory. CC: received physical exercise materials but did not participate in a cognitive or exercise programme. | Processing speed (computerised modified Stroop). | IG1, IG2, and IG3 improved executive function performance, but only IG1 showed a general facilitative effect on nonexecutive control. |
| [42] (2018) | 13 F + 13 M (aged mean 74.24 years old). | Verified if 6-week exercise training improved physical fitness in gait speed and cognition in the memory domain in older adults. | Retired professors (education for 16–19 years) from Beijing, China. | 6 weeks; aerobic (dance training exercise program); 4×/week; 45 min exercise. | Verbal memory (RAVLT), verbal episodic memory (WMS—LM), working memory (WAIS-III Digit-Symbol Substitution Modality Test and Digit Span), executive function (TMT), visual memory (BVMT). | After the exercise program, participants showed significantly improved memory performances in the Logical Memory Test (WMS—LM) and Rey Auditory Verbal Learning Test. The WAIS-III Digit Span showed a marginally significant increase. |
| [43] (2022) | 33 F + 7 M (IG = 20, aged mean 67.5 years old; CG = 20, aged mean 67.6 years old). | Determined the effect of 12-week Judo training on cognitive processing and muscle function among the elderly. | Health Promotion Center in Poland. | IG: 12 weeks; strength and resistance (Judo); 3×/week; 45 min of exercise. CG: did not undertake any exercises during the experiment. | Processing speed (Stroop Test) | IG-improved Stroop performance reflected by shortening the response time related to the Stroop "naming" interference. |

**Table 2.** *Cont.*

| Authors and Year | Sample Characteristics | Purpose | Setting and Country | Intervention (Duration; Main Characteristics of the Program) | Cognitive Outcome (Measures) | Main Results |
|---|---|---|---|---|---|---|
| [44] (2022) | 51 F + 4 M (IG1 = 28, aged mean 67.7 years old; IG2 = 27, aged mean 64.6 years old). | Analysed the influence of sitting callisthenic balance and resistance training on cognitive function and the mediating role of the change in the level of neurotrophic factors and strength in older, healthy participants. | Community in Bydgoszcz, Poland. | IG1 (sitting callisthenic balance): 3 months; stretching, mobility exercises, basic core-strength, and balance exercises (without additional weights); 2×/week; 45 min of exercise. IG2 (resistance training): 3 months; strength and resistance (machine and non-machine-based exercises); 2×/week; 50 min of exercise. | Global cognitive function (MoCA), cognitive function (Sprawnosci Operacyjnej Test), executive function (TMT B), auditory attention (DST), working memory (DST Backwards). | Both IG1 and IG2 influenced multiple cognitive domains. The IG2 program improved global cognitive functioning, decision making, and visual attention. In IG1, set shifting and the short-term visual memory processing speed of simple. visual stimuli were improved. |
| [45] (2022) | 26 F (IG = 13, aged mean 62.5 years old; CG = 13, aged mean 69.5 years old). | Assessed the effects of 4-movement Qigong on cognitive function and physical performance. | Community in Innsbruck, Austria. | IG: 8 weeks; function and flexibility (4-movement Qigong); 3×/week; 45 min of exercise. CG: maintained their usual activities of daily living throughout the study period. | Executive function (TMT-A and TMT-B); memory function (DST Forwards and Backwards). | IG significantly improved TMT-A and TMT-B. |
| [46] (2021) | 31 F + 15 M (IG1 = 25, aged mean 79.6 years old; IG2 = 21, aged mean 83.8 years old). | Compared the effects of Kinect-based exergaming and combined physical exercise training on cognitive function and brain activation in frail, older adults. | Care centres for communities in Taiwan, China. | IG1: exergames, 12 weeks; resistance (upper and lower extremity movements using 3D Space), aerobic (swimming and running in a 3D space), and balance (balance exercises and Tai Chi) training; 3×/week; 60 min exercise. IG2: physical exercise, 12 weeks; resistance (upper and lower extremity movements using Theraband), aerobic (stepping variations), and balance (balance exercises and Tai Chi) training; 3×/week; 60 min exercise. | Global cognition (MoCA), executive function (EXIT 25), verbal memory (CCVLT), attention (CWT), working memory (spatial n-back task test). | Both groups improved significantly in global cognition, executive function, and attention after the 12-week intervention. Only the IG1 group showed significant improvements in verbal and working memory after the intervention. |

**Table 2.** *Cont.*

| Authors and Year | Sample Characteristics | Purpose | Setting and Country | Intervention (Duration; Main Characteristics of the Program) | Cognitive Outcome (Measures) | Main Results |
|---|---|---|---|---|---|---|
| [47] (2021) | 14 F + 6 M (aged mean 69.1 years old). | Determined whether a 12-week strength training program could improve fluid cognition in healthy, older adults and explore concomitant physiological and psychological changes. | Community in Los Angeles, USA. | 12 weeks; resistance training (total body strength using machine-based exercises); 3×/week; 60 min exercise. | NIHTB-CB fluid cognition (inhibitory control and attention, episodic memory, working memory, executive function, and processing speed). NIHTB-CB crystallised cognition (vocabulary and reading recognition). | NIHTB-CB fluid composite scores significantly increased from pre- to post-interventions, while NIHTB-CB crystallised composite scores did not. Performances on individual fluid instruments, including executive function, attention, working memory, and processing speed, also significantly improved. |
| [48] (2017) | 25 F + 28 M (IG = 29, aged mean 73.3 years old; CG = 24, aged mean 77 years old). | Understood the effect of aerobic training on cerebral metabolism and cerebral grey matter volume in older adults. | Assisted living facilities in Frankfurt am Main, Germany. | IG: 12 weeks; aerobic (bicycle); 3×/week; 30 min of exercise. | Memory (VLMT early recall, VLMT late recall, VLMT recognition, and CERAD figure recall), executive control (Stroop interference), working memory (Digit span forwards, Digit span backwards), verbal fluency (semantic fluency and phonematic fluency). | The analysis did not reveal a significant effect of any cognitive domains assessed in this study. |
| [49] (2021) | 148 F + 49 M (females were 75% of the participants and were considered people ≥ 60 years old). | Observed if the Healthy Ageing Promotion Program for You, promoted improvements in cognition, frailty status, functional status, perceived health, and reduction in social isolation. | Community in Singapore. | 3 months; dual-task exercises (resistance, balance, aerobic, and cognitive tasks); 1 or 2×/week; 60 min exercise. | Global cognition (MoCa). | There was significant improvement in the MoCA scores at 3 months for the entire group. |
| [50] (2023) | 24 F + 62 M (IG1 = 27, aged mean 67.9 years old; IG2 = 30, aged mean 67.2 years old; CG = 29, aged mean 68.3 years old). | Analysed the effects of two short-term aerobic exercises on cognitive function in healthy, older adults during COVID-19. | Home in Tokyo and Kanagawa, Japan. | IG1 (walking group): 4 weeks; aerobic (walking with Nordic ski poles in both hands) 3×/week; 30 min of exercise. IG2 (dance): 4 weeks; aerobic (dance) 3×/week; 30 min of exercise. CG: maintained their usual activities of daily living throughout the study period. All groups consumed amino acid-containing foods 3×/week. | Global cognitive functioning (MoCA); executive function (FAB). | The results show that both exercise intervention groups (IG1 and IG2) improved executive function, while the dance group (IG2) showed an additional improvement in global cognitive function. |

**Table 2.** *Cont.*

| Authors and Year | Sample Characteristics | Purpose | Setting and Country | Intervention (Duration; Main Characteristics of the Program) | Cognitive Outcome (Measures) | Main Results |
|---|---|---|---|---|---|---|
| [51] (2018) | 17 F + 2 M (IG = 8, aged mean 75.0 years old; CG = 11, aged mean 71.9 years old). | Examined the effect of 2-year cognitive-motor dual-task training on the cognitive function and motor ability of healthy, elderly people without marked cognitive impairments. | Community dwellers in Sumiyoshi-ku, Osaka City, Japan. | IG: 2 years; mental gymnastics, resistance training, aerobic exercise, and flexibility exercises; 1×/week; 60min of exercise. | Registration and recall, long-term memory, orientation, attention, verbal fluency and understanding, word retrieval, visuospatial skills, abstract meaning (Modified MMSE—3MS). Proc. Speed (TMT). | Participation in IG maintained the scores in almost all domains of cognitive function, as well as the total 3MS scores. 3MS scores decreased in CG. |
| [52] (2017) | 10 F + 12 M (IG1 = 12, aged mean 68.25 years old; IG2 = 10, aged mean 68.6 years old). | Assessed whether a dance training program that stressed the constant learning of new movement patterns was superior in terms of neuroplasticity to conventional fitness activities, | Community in Germany. | First 6 months: IG1: aerobic training (dance), 2×/week; 90 min exercise. IG2: strength and endurance training (cycle ergometer and exercises for all body parts), 2×/week; 90 min exercise. Last 12 months: same training for both groups, but only 1×/week. | Verbal short- and long-term memory (RAVLT); attention (TAP) | Both groups showed significant improvements in attention after 6 months and verbal memory after 18 months. In terms of cognitive ability, no group differences emerged. |
| [53] (2022) | 16 F + 24 M (IG1 = 20, aged mean 64.05 years old; IG2 = 20, aged mean 65.50 years old). | Compared the efficiency of dual-task training versus aerobic exercise training in improving cognitive function in healthy, older individuals. | Community in Wardha, Maharash-tra, India. | IG: 6 weeks; dual-task training (cognitive + motor activities); 3×/week; 45 min. IG2: 6 weeks; aerobic training (treadmill, bicycle, and walking exercises); 5×/week; 45 min. | Working memory (TMT-A), executive Function (TMT-B), general global cognitive function (MoCA). | Both groups noted post-intervention improvements in TMT-A, TMT-B, and MoCA scores. However, the difference was more significant for group IG1 than group IG2. |
| [54] (2021) | 17 F + 16 M (IG = 17, aged mean 77.35 years old; CG = 16, aged mean 76.81 years old). | Investigated the effect of an intervention combining exercise and cognitive activity on cognitive function in healthy, older adults. | Community in Kobe city, Japan. | IG: 3 mounts, stretching, muscle strength, and dual-task aerobic exercises, 1×/week; 50 min exercise. Also had a homework cognitive task twice a week. CG: no received intervention. | Cognitive impairment (MMSE), long- and short-term memory (Logical Memory IIA), executive function (TMT), motor speed, attention, visual perception (DSST). | A significant improvement in long-term and short-term memory was observed in the IG at follow-up and five or six months after the conclusion of the intervention. Additionally, the decline in MMSE scores in the IG was lower than in the CG. |

**Table 2.** *Cont.*

| Authors and Year | Sample Characteristics | Purpose | Setting and Country | Intervention (Duration; Main Characteristics of the Program) | Cognitive Outcome (Measures) | Main Results |
|---|---|---|---|---|---|---|
| [55] (2017) | 15 F + 14 M (IG = 14, aged mean 69.7 years old; CG = 15, aged mean 68.6 years old). | Investigated the effects of exergame training over 6 weeks on healthy, old participants' cognitive, motor, and sensory functions. | Community in Leipzig, Germany. | IG: exergames 6 weeks; strength (hurdles, javelin throwing, 100 m running), aerobic (swimming, hammer throwing), and coordination (trampoline, high diving, archery, mountain biking), 2×/week; 60 min exercise. IG2: no received intervention | Attention (TAP 2.3), working memory (N-back task), alertness and simple reaction time (reaction time to a visual stimuli), response inhibition (response triggered by an external stimulus). | IG showed significant performance improvements in alertness and simple reaction time from baseline to post-measurement. These improvements did not result in differential performance improvements when comparing IG and CG. |
| [56] (2023) | 19 F + 16 M (IG1 = 7, aged mean 63.7 years old; IG2 = 11, aged mean 63.1 years old; IG3 = 17, aged mean 64.6 years old). | Investigated the effects of simultaneous exercise and cognitive training on several cognitive domains in healthy, older adults compared to training alone. | Does not present this information. | IG1: exercise training: 24 weeks; aerobic (stationary bicycle); 2×/week; 30 min of exercise. IG2: cognitive training: 24 weeks; cognitive games; 2×/week; 30 min of exercise. IG3: exercise + cognitive training. Combine IG1 and IG2 interventions. | Executive functions (TMT and Computerised Modified Stroop task), verbal memory (Rey Words test), working memory (2-back test). | All groups improved their executive performances, including flexibility or working memory. Simultaneous exercise and cognitive training ere more efficient than either training alone to improve executive function. |
| [21] (2020) | 57 F + 27 M (IG1 = 20, aged mean 67.6 years old; IG2 = 21, aged mean 66.3 years old; IG3 = 19, aged mean 68.1 years old; IG4 = 14, aged mean 69.2 years old). | Evaluated the effects of simultaneous aerobic exercise and cognitive training intervention on dual-task walking performance on healthy, older adults. | Community, in Tucson, Arizona, USA. | IG1: aerobic exercise and cognitive training, combined IG2 and IG3 conditions described below. IG2: cognitive training, 12 weeks; cognitive training, 3×/week; 30 min exercise. IG3: aerobic exercise, 12 weeks; aerobic training (stationary recumbent bicycle), 3×/week; progressively increased exercise from 15 to 30 min. IG4: video-watching control, 12 weeks; watching videos, 3×/week; 30 min. | Serial subtraction during two-minute walk (DTWT) | IG1, IG2, and IG3 groups significantly improved in the cognitive aspect of the DTWT following the full 12-week intervention. The improvements in IG1 were twice as much as in the other groups and were significant at 6 weeks. |
| [57] (2018) | 28 F + 9 M (IG1 = 18, aged mean 67.6 years old; IG2 = 19, aged mean 69.1 years old). | Compared the effects of Poi and Tai Chi on physical and cognitive functions in healthy, older adults. | Does not present this information. | IG1: Poi (a weight on the end of a cord that is swung in circular patterns around the body), 4 weeks, Poi exercise (strength/functional), 2×/week, 60 min. IG2: Tai Chi , 4 weeks, Tai Chi exercise, 2×/week, 60 min. | Verbal memory and visual memory, finger tapping, symbol digit coding, Stroop test, shifting attention, and continuous performance (CNS Vital Signs) | Both groups increased simple attention, complex attention, cognitive flexibility, psychomotor speed, and executive function. Both groups declined in composite memory and visual memory tests. |

**Table 2.** *Cont.*

| Authors and Year | Sample Characteristics | Purpose | Setting and Country | Intervention (Duration; Main Characteristics of the Program) | Cognitive Outcome (Measures) | Main Results |
|---|---|---|---|---|---|---|
| [58] (2020) | 29 F + 17 M (IG = 23, aged mean 65.5 years old; CG = 23, aged mean 67.7 years old). | Compared the effects of 12 weeks of aerobic training versus control condition on cardiorespiratory fitness, cognition, and magnetic resonance imaging. | Community in Oxford, England. | IG: 12 weeks, aerobic exercise (cycling), 3×/week, 30 min. CG: continued with their normal routines and did not begin a PA programme. | Executive function (TMT B and COGSTATE Two-Back), memory (HVLT-R and RCF), processing speed (TMT A. Digit Coding and CNTABRTTT). | There were no significant differences in cognitive measures between the aerobic training and control groups. |
| [59] (2018) | 90 F + 15 M (IG = 60, aged mean 73.59 years old; CG = 45, aged mean 73.22 years old). | Examined whether Bingocize (Game Centerer Mobile Application) could improve aspects of physical and cognitive performances. | Senior centres in Kentucky and Tennessee, USA. | IG: Bingo + health education + exercise, 10 weeks, 12 exercises of cardiovascular, strength, balance, and flexibility exercises, 2×/week; 60 min CG: Bingo + health education, 10 weeks, 2×/week; 60 min. | Monitoring and adjusting working memory contents, switching flexibly between tasks, and deliberately overriding dominant/prepotent responses (EXAMINER cognitive battery). | IG performed better than the control group in the "updating" (executive function) task of the EXAMINER battery. However, this was subsumed by a significant interaction. |
| [60] (2017) | 64 M (IG1 = 22, aged mean 66.88 years old; IG2 = 21, aged mean 66.15 years old; IG3 = 21, aged mean 65.7 years old). | Explored the effects of 6-month open- and closed-skill exercise interventions on the neurocognitive performance of the elderly when performing the task-switching paradigm and N-back task. | Community in Taiwan. | IG1: 24 weeks, skill training (table-tennis games), 3×/week, 40 min). IG2: 24 weeks, aerobic exercise (treadmill or cycling), 3×/week, 40 min. IG3: 24 weeks, balance and stretching, 3×/week, 40 min. | Switching capacity (task-switching paradigm); working memory (N-back task). | IG1 and IG2 did not increase their accuracy rates in the task-switching paradigm but presented significantly faster responses than IG3 in the switch trials. In terms of the N-back task, the two exercise groups significantly increased their accuracy rates in the 1-back condition, and IG1 also showed an improvement in the accuracy rate in the 2-back condition. |
| [61] (2023) | 63 F + 19 M (IG1 = 27, aged mean 72.1 years old; IG2 = 29, aged mean 73.3 years old; CG = 26, aged mean 72.1 years old). | Compared the effects of a supported yoga-based exercise intervention on verbal fluency to an aerobic exercise intervention and a wait-list control group. | Community in Stockholm, Sweden. | IG1 (yoga): 12 weeks; yoga; ≥3×/week; 60 min of exercise. IG2 (aerobic): 12 weeks; aerobic (cycling/spinning, dance-based exercise) ≥3×/week; 60 min of exercise. CG: maintained their usual activities of daily living throughout the study period. | Verbal fluency (total FAS, animals, and verbs) | Participation in yoga or aerobic exercises was associated with estimated improvements in verbal fluency compared to a non-active control group. |

**Table 2.** *Cont.*

| Authors and Year | Sample Characteristics | Purpose | Setting and Country | Intervention (Duration; Main Characteristics of the Program) | Cognitive Outcome (Measures) | Main Results |
|---|---|---|---|---|---|---|
| [62] (2020) | 12 F + 14 M (IG = 15, aged mean 69.7 years old; CG = 9, aged mean 71.9 years old). | Observed the effect of dual-task exercise (Synapsology) to improve physical and cognitive functions. | Community in Tsukuba, Japan. | IG: 8 weeks, dual-task exercises (stretching, lower body exercises, and walking + cognitive tasks), 2×/ week; 60 min. CG: no received intervention. | Executive function (TMPT, 25-hole peg test). | Cognitive function results in the 25-hole trail-making peg test had a statistically significant difference within the IG. However, the differences in the TMPT between the IG and CG were insignificant. |
| [63] (2022) | 24 F + 14M (IG = 13, aged mean 62.5 years old; CG = 13, aged mean 69.5 years old). | Explored the effects of an active videogame intervention on fitness and cognitive functions in older adults. | Community in Xian Province and Shaanxi Province, China. | IG (videogame exercises): 12 weeks; aerobic (Zumba, aerobic boxing, and virtual tennis); 3×/week; 50–55 min of exercise. CG: maintained their usual activities of daily living throughout the study period. | Processing speed, spatial ability, working memory, language ability, and associative memory (specific software) | The results show improvements in cognition (spatial cognition) in the IG. |

Notes: M (Male), F (Female), IG (Intervention Group), CG (Control Group), MMSE (Mini-Mental State Examination), VCAP (Virginia Cognitive Aging Project), DRS-2 (Mattis Dementia Rating Scale 2), RAVLT (Rey Auditory Verbal Learning Test), COWAT (Phonemic Controlled Oral Word Association Test), ACE-R (Addenbrooke's Cognitive Examination Revised), CASI 2.0 (Cognitive Abilities Screening Instrument 2.0), CDR (Clinical Dementia Rating), FUCAS (Functional Cognitive Assessment Scale), TEA (Test of Everyday Attention), TMT (Trail-Making Test), ROCF (Rey–Osterrieth Complex Figure Test), RBMT (Rivermead Behavioral Memory Test), VFT (Verbal Fluency Test), DSST (Digit Symbol Substitution Test), DMT (Digit Memory Test), Stroop (Stroop Colour and Word Test), MoCA (Montreal Cognitive Assessment), COGTEL (Cognitive Telephone Screening Instrument), CANTAB (Cambridge Neuropsychological Test Automated Battery), WMS—LM (Chinese Version of Logical Memory Subtest of the Wechsler Memory Scale), BVMT (Benton Visual Retention Test), EXIT 25 (Executive Interview 25), CCVLT (California Verbal Learning Test), CWT (Stroop Colour and Word Test), VLMT (Verbal Learning and Memory Test), CERAD (Consortium to Establish a Registry for Alzheimer's Disease), TAP (Test of Attentional Performance), DTWT (Dual-Task Walking Test), TMPT (Trail-Marking Peg Test), HVLT-R (Hopkins Verbal Learning Test Revised), CNTABRTTT (Cambridge Neuropsychological Test Automated Battery Reaction-Time Touchscreen Task), VST (Victoria Stroop Test), SPMT (Scenery Picture Memory Test), DST (Digit Span Test), FAB (Frontal Assessment Battery at Bedside); * participants with mild cognitive impairments were not considered for analyses.

### 2.5. Study Quality and Risk of Bias

The Quality Assessment Tool for Quantitative Studies of the Effective Public Health Practice Project (EPHP) Field [64] was used to assess the study quality and risk of bias. This instrument assesses six components, which are (1) selection of bias; (2) study design; (3) confounders parameter that evaluates the differences that exist between the groups before the intervention; (4) the "blindness" of the outcome assessors and the participants regarding the awareness of the intervention exposure status and research question, respectively; (5) methods/instruments used to collect the data; and (6) the withdrawal report. Subsequently, each study achieved a final score according to the instrument rules. This information is presented in Table 3 and was independently assessed by the three authors (M.A.T., F.S., and S.M.). Any differences were analysed and were resolved in agreement.

**Table 3.** Study methodological quality assessment using the EPHPP.

| Authors | Selection Bias | Design | Confounders | Blinding | Data Collection Methods | Withdrawals and Drop-Outs | Overall |
|---|---|---|---|---|---|---|---|
| [27] | Fair | Good | Good | Poor | Good | Good | Moderate |
| [28] | Fair | Good | Good | Poor | Good | Good | Moderate |
| [29] | Good | Good | Good | Good | Good | Good | Strong |
| [30] | Fair | Fair | Good | Poor | Good | Good | Moderate |
| [31] | Fair | Good | Good | Good | Good | Good | Strong |
| [32] | Fair | Good | Good | Poor | Good | Good | Moderate |
| [33] | Poor | Good | Good | Poor | Good | Good | Weak |
| [34] | Poor | Fair | Good | Poor | Good | Poor | Weak |
| [35] | Fair | Good | Good | Good | Good | Good | Strong |
| [36] | Good | Good | Good | Fair | Good | Fair | Strong |
| [37] | Fair | Good | Good | Fair | Good | Good | Strong |
| [38] | Good | Fair | N/a | Poor | Good | Good | Moderate |
| [39] | Poor | Good | Good | Good | Good | Good | Moderate |
| [40] | Good | Fair | Good | Poor | Good | Fair | Moderate |
| [42] | Poor | Fair | N/a | Poor | Good | Good | Weak |
| [41] | Good | Good | Good | Poor | Good | Good | Moderate |
| [43] | Poor | Good | Good | Poor | Good | Good | Weak |
| [44] | Good | Good | Good | Fair | Good | Fair | Strong |
| [45] | Fair | Good | Good | Poor | Good | Good | Moderate |
| [46] | Fair | Good | Good | Fair | Good | Fair | Strong |
| [47] | Poor | Fair | N/a | Poor | Good | Good | Weak |
| [48] | Good | Good | Good | Fair | Good | Good | Strong |
| [49] | Good | Fair | N/a | Poor | Good | Fair | Moderate |
| [50] | Good | Good | Good | Fair | Good | Good | Strong |
| [51] | Fair | Good | Good | Poor | Good | Good | Moderate |
| [53] | Poor | Good | Good | Fair | Good | Good | Moderate |
| [52] | Fair | Good | Good | Poor | Good | Poor | Weak |
| [54] | Fair | Good | Good | Fair | Good | Good | Strong |
| [55] | Poor | Good | Good | Poor | Good | Good | Weak |
| [56] | Poor | Good | Good | Poor | Good | Poor | Weak |
| [21] | Fair | Good | Good | Fair | Good | Fair | Strong |
| [57] | Poor | Good | Good | Fair | Good | Good | Moderate |
| [58] | Poor | Good | Good | Fair | Good | Good | Moderate |
| [59] | Good | Good | Good | Good | Good | Good | Strong |
| [60] | Fair | Fair | Good | Fair | Good | Good | Strong |
| [61] | Good | Good | Good | Fair | Good | Good | Strong |
| [62] | Fair | Good | Good | Poor | Good | Good | Moderate |
| [63] | Good | Good | Good | Poor | Good | Good | Moderate |

Note. N/a (Not applicable).

## 3. Results

The flowchart is presented in Figure 1. In the identification phase, 1922 articles were found in the database search. Of these articles, 375 were duplicates, and after their elimination, 1547 were recorded for the title and abstract screening process. In this phase, 1418 articles were removed because they did not meet the eligibility criteria. Thus, 129 remained for a full review. Ninety-one articles were deleted for reasons related to (1) the intervention type, such as not having a physical fitness program, not being an observational or experimental study, or presenting other types of outcomes (n = 57); (2) the population, when the sample had some type of disease and/or was younger than 60 years old (n = 25); (3) not being able to find the articles (n = 5); (4) being written in another language (n = 1); and (5) articles that only presented the intervention protocol but not the final results (n = 3). Therefore, 38 studies were considered as relevant for inclusion.

### 3.1. Study Quality and Risk of Bias

Due to the methodological quality analysis performed on the articles included (Table 3), fourteen studies were classified as strong [21,28,31,35–37,44,46,48,50,54,59–61], sixteen moderate [27,28,30,32,38–41,45,49,51,53,57,58,62,63], and eight weak [33,34,42,43,47,52,55,56]. Regarding the individual parameters analysed, the following actions were performed. (1) In the selection of bias, twelve studies were classified as good [29,36,38,40,41,44,48–50,59,61,63], because the sample was very likely to be representative of the target population and have a rate equal to or higher than 80% in terms of participation. (2) In the study design, a good score was assigned when the articles were randomised controlled [21,27–29,31,35–37, 39,44,46,48,50,54,57,60,61,63] or controlled trials [32,33,41,43,45,51–53,55,56,58,59,62] and fair [30,34,38,40,42,47,49,60] when they presented another type of design. (3) Regarding the confounder parameter, which evaluated the differences between the analysed groups, thirty four studies were rated as strong, since they had no difference between the groups or were controlled for a least 80% of the relevant confounders. Studies that only analysed one group (n = 4) [38,42,47,49] were not classified. (4) In the blinding process, if the assessor was not aware of the intervention status of participants and the study participants were not aware of the research question, the study was classified as good (n = 5) [29,31,35,39,59]; when studies met only one of the abovementioned conditions, they were classified as fair (n = 13) [21,36,37,44,46,48,50,53,54,57,58,60,61], and when they did not meet any, they were considered poor (n = 20) [27,28,30,32–34,38,40–43,45,47,49,51,52,55,56,62,63]. (5) In terms of the data collection methods, all the studies presented collection tools that were valid and reliable, so were classified as strong. (6) The last points assessed were the withdrawals and drop-outs; studies with a follow-up rate $\geq$ 80% were classified as good (n = 29) [27,28,30–33,35,37,38,41–43,45,47,48,50,51,53–55,57–63,65], between 60% and 79% were fair (n = 6) [21,36,40,44,46,49], and less than 60% were poor (n = 3) [34,52,56].

### 3.2. Intervention Characteristics

The pertinent information of all the included studies is systematised in Table 2. Regarding the main characteristics of the interventions, the present review presented a total of 2389 older adults analysed. Most participants were female (n = 1515). In terms of age, the studies covered an average age between 60 and 84 years old. Two studies [30,34] contained groups that had mild cognitive impairments, but such groups were not considered. Only healthy groups from those two studies were analysed.

The type of intervention program most observed was aerobic training sessions (n = 19) [21,27,28,30,33,34,36,38,41,42,48,50,52,53,56,58,60,61,63]. Dance was the most used methodology in this type of intervention (n = 9) [28,33,34,36,42,50,52,61,63], but other forms of training physical capacity were also used, such as walking on the treadmill (n = 7) [28,30,33,38,41,50,53] or stationary cycling (n = 8) [21,36,48,52,53,56,58,60]. Still considering these aerobic training studies, for eleven of them [21,28,33,36,41,50,52,53,56,60,61], the difference between aerobic training and another type of training (i.e., stretching, strength, balance, etc.) was analysed across various intervention groups. Other studies (n = 9) adopted resistance, strength, or functional training sessions performed with or without using exercise machines or other accessories [32,35,37,39,43–45,47,57]. In addition, ten studies [29,31,40,46,49,51,54,55,59,62] adopted a mixed methodology, performing aerobic, strength, flexibility, or balance training sessions in a combined way. Innovative methodologies were also used for the performance of physical exercise based on exergames (n = 4) [37,46,55,63].

Most of the interventions had a duration of three months (n = 19) [21,27,30,31,36,37,39–41,43,44,46–49,54,58,61,63] and only ten had a longer duration [28,29,32–34,38,51,52,56,60], highlighting the studies of Muller, Rehfeld, Schmicker, Hokelmann, Dordevic, Lessmann, Brigadski, Kaufmann, and Muller [52] which lasted one year, and Morita, Yokoyama, Imai, Takeda, Ota, Kawai, Suzuki, and Okazaki [51] that had a duration of 2 years. The intervention with the shortest period was only four weeks [57].

In terms of weekly frequency training, twice and three times a week were the most common. Only the studies by Morita, Yokoyama, Imai, Takeda, Ota, Kawai, Suzuki, and Okazaki [51], and Murata, Ono, Yasuda, Tanemura, Kido, and Kowa [54] had sessions once a week; on the other hand, the study by Ji, Pearlson, Zhang, Steffens, Ji, Guo, and Wang [42] had a frequency of 4 times a week. One study did not have this feature of weekly frequency clarified, as the participants were encouraged to perform training sessions at home [39].

### 3.3. Main Results

Table 4 summarises the main results of each study, including information related to the CF (instruments/tasks, outcomes evaluated, and the effects of the interventions). The main outcomes were organised into three main categories: the first two were EF and memory function (MF) domains, as presented in the review by de Asteasu, Martinez-Velilla, Zambom-Ferraresi, Casas-Herrero, and Izquierdo [6], and the third one was global performance (GP). The EF subdomains were working memory, attention, verbal fluency, reasoning, and processing speed. On the other hand, the MF subdomains were recognition, immediate recall, delayed recall, facial name recall, and paired associations. The GP category included studies that used a cognitive assessment instrument that produced a final score, such as the Montreal Cognitive Assessment (MoCA), COGTEL, or the Mini-Mental State Examination (MMSE).

**Table 4.** Cognitive tasks assessing outcomes and main effects of interventions on cognitive function.

| Authors | Cognitive Instruments Used | Outcomes | Main Categories | Effects of the Intervention on Executive Function Domains | Effects of the Intervention on Memory Domains | Effects of the Intervention on Global Performance |
|---|---|---|---|---|---|---|
| [27] | MMSE | Global performance (final score) | GP | N/a | N/a | Not improved |
| [28] | (1) VCAP; (2) Switching Task and Spatial Working Memory Task | Proc. speed, vocabulary, fluid reasoning, and working memory | EF | Improved | N/a | N/a |
| [29] | (1) ZAVEN; (2) Letter Fluency Test; (3) TMT; (4) VST and CVLT; (5) TAP; (6) Logical Memory Task; (7) DSST | Global performance, executive function, memory and proc. Speed | GP, EF, MF | Improved | Not improved | Improved |
| [30] | (1) RAVLT; (2) COWAT; (3) Animal Fluency Test | Verbal memory and verbal fluency | EF and MF | Improved | Improved | N/a |
| [31] | ACE-R | Attention, memory, verbal fluency, language and visual spatial skills | EF and MF | Improved | Improved | N/a |
| [32] | (1) VFT; (2) TUG + Cognitive Test; (3) SPMT; (4) TMT | Verbal fluency, short-term memory, executive function | EF and MF | Improved | Not improved | N/a |
| [33] | CASI 2.0 | Global performance score | GP | N/a | N/a | Improved |
| [34] | (1) CDR; (2) FUCAS; (3) TEA; (4) TMT; (5) RAVLT; (6) VFT | Attention, memory, orientation, verbal fluency | EF and MF | Improved | Improved | N/a |
| [35] | (1) DSST; (2) DMT; (3) Stroop; (4) TMT | Proc. speed, working memory, visuospatial processing, attention | EF | Improved | N/a | N/a |
| [36] | (1) Dual Task; (2) N-Back Task; (3) Digit Stroop Task; (4) MoCA | Working memory, attention, proc. speed, reaction time, global performance score (MoCA) | EF and GP | Improved | N/a | Not improved |

**Table 4.** *Cont.*

| Authors | Cognitive Instruments Used | Outcomes | Main Categories | Effects of the Intervention on Executive Function Domains | Effects of the Intervention on Memory Domains | Effects of the Intervention on Global Performance |
|---|---|---|---|---|---|---|
| [37] | COGTEL | Prospective memory, verbal short-term memory, long-term memory, working memory, verbal fluency, inductive reasoning | GP, EF, MF | Not improved | Improved | Improved |
| [38] | (1) Symbol Digit Modalities Test; (2) Cart Sorting Test; (3) Colour and Word Inference Test; (4) Buschke Selective Reminding Test; (5) MCGCF; (6) VFT; (7) Memory (Medical College of Georgia Complex Figure); (8) Auditory Consonant Trigram Test | Proc. speed, verbal memory, visual memory, verbal fluency and attention | EF and MF | Improved | Improved | N/a |
| [39] | (1) MoCA; (2) Flanker Task Test | Global performance score (MoCA) and proc. speed | EF and GP | Improved | N/a | Improved |
| [40] | (1) Motor Screening Test; (2) PAL Test; (3) RVT Test | Episodic memory, visual attention, proc. speed | EF and MF | Improved | Improved | N/a |
| [42] | (1) RAVLT; (2) WMS—LM; (3) WAIS III Digit Span; (4) TMT—A and BVMT | Verbal memory, verbal episodic memory, working memory, visual memory | EF and MF | Marginals improvements | Improved | N/a |
| [41] | Computerised Modified Stroop Test | Proc. speed | EF | Improved | N/a | N/a |
| [43] | Stroop Test | Proc. speed | EF | Improved | N/a | N/a |
| [44] | (1) MoCA; (2) Sprawnosci Operacyjnej Test; (3) TMT B; (4) DST and DST Backwards | Global performance, executive function, auditory attention, working memory | EF, MF, GP | Improved | Improved | Improved |
| [45] | (1) TMT-A and TMT-B; (2) DST Forwards and Backwards | Executive function and memory function | EF and MF | Improved | Not improved | N/a |
| [46] | (1) MoCA; (2) EXIT 25; (3) CCVLT; (4) CWT; (5) Spacial N-Back Task Test | Global performance score (MoCA), verbal memory, attention, working memory | EF, MF, GP | Improved | Improved | Improved |
| [47] | (1) NIHTB-CB Fluid Cognition; (2) NIHTB-CB Crystallised Cognition | Attention, working memory, episodic memory proc. speed, vocabulary and reading recognition | EF and MF | Improved | Not improved | N/a |
| [48] | (1) VLMT; (2) CERAD; (3) STROOP Interference; (4) Digit Span—Forwards and Backwards; (5) Semantic Fluency Test | Memory recall, executive control, working memory, and verbal fluency | EF and MF | Not improved | Not improved | N/a |
| [49] | MoCa | Global performance score | GP | N/a | N/a | Improved |
| [50] | (1) MoCA; (2) FAB | Global performance and executive function | EF and GP | Improved | N/a | Improved |
| [51] | (1) Modified MMSE—3MS; (2) TMT | Registration and recall, long-term memory, orientation, attention, verbal fluency and understanding, word retrieval, visuospatial skills, abstract meaning, proc. speed | EF and MF | Not improved | Not improved | N/a |

**Table 4.** *Cont.*

| Authors | Cognitive Instruments Used | Outcomes | Main Categories | Effects of the Intervention on Executive Function Domains | Effects of the Intervention on Memory Domains | Effects of the Intervention on Global Performance |
|---------|----------------------------|----------|-----------------|-----------------------------------------------------------|-----------------------------------------------|---------------------------------------------------|
| [52] | (1) RAVLT; (2) TAP | Short- and long-term memory and attention | EF and MF | Improved | Improved | N/a |
| [53] | (1) TMT-A; (2) TMT-B; (3) MoCA | Global performance, executive function and working memory | EF and GP | Improved | N/a | Improved |
| [54] | (1) MMSE; (2) Logical Memory IIA; (3) TMT; (4) DSST | Global performance score (MMSE), long- and short-term memory, attention, proc. speed | EF, MF, GP | Not improved | Improved | Not improved |
| [55] | (1) TAP 2.3; (2) N-Back Task; (3) Simple Reaction-Time Task; (4) Response Inhibition Talk | Attention, working memory, proc. speed, reaction time, response inhibition | EF | Improved | N/a | N/a |
| [56] | (1) TMT; (2) Computerised Modified Stroop Task; (3) Rey Words Test; (4) 2-Back Test | Executive function, verbal memory, working memory | EF and MF | Improved | Not improved | N/a |
| [21] | Serial Subtraction during a Two-Minute Walk | Working memory and proc. speed | EF | Improved | N/a | N/a |
| [57] | CNS Vital Signs | Verbal memory, visual memory, proc. speed, attention and reaction times | EF and MF | Improved | Not improved | N/a |
| [58] | (1) TMT—A and B; (2) GOGSTATE Two Back: (3) HVLT-R; (4) ROCF; (5) CNTABRTTT | Attention, working memory, proc. speed, reaction time, memory | EF and MF | Not improved | Not improved | N/a |
| [59] | EXAMINER Cognitive Battery—EF Domains | Working memory, switching flexibly between tasks, deliberately overriding dominant/prepotent responses | EF | Improved | N/a | N/a |
| [60] | (1) Task-Switching Paradigm; (2) N-Back Test | Working memory and proc. speed | EF | Improved | N/a | N/a |
| [61] | Total FAS, Animals, and Verbs | Verbal fluency | EF | Improved | N/a | N/a |
| [62] | TMPT—25-Hole Peg Test | Proc. speed and visual attention | EF | Improved | N/a | N/a |
| [63] | Tasks on a Specific Software | Processing speed, spatial ability, working memory, language ability, associative memory | EF and MF | Improved | Not improved | N/a |

Notes. N/a (Not applied), EF, (Executive Function), MF (Memory Function), GP (Global Performance), MMSE (Mini-Mental State Examination), VCAP (Virginia Cognitive Aging Project), RAVLT (Rey Auditory Verbal Learning Test), COWAT (Phonemic Controlled Oral Word Association Test), ACE-R (Addenbrooke's Cognitive Examination Revised), CASI 2.0 (Cognitive Abilities Screening Instrument 2.0), CDR (Clinical Dementia Rating), FUCAS (Functional Cognitive Assessment Scale), TEA (Test of Everyday Attention), TMT (Trail-Making Test), ROCF (Rey–Osterrieth Complex Figure Test), VFT (Verbal Fluency Test), DSST (Digit Symbol Substitution Test), DMT (Digit Memory Test), Stroop (Stroop Colour and Word Test), MoCA (Montreal Cognitive Assessment), COGTEL (Cognitive Telephone Screening Instrument), WMS—LM (Chinese Version of Logical Memory Subtest of the Wechsler Memory Scale), BVMT (Benton Visual Retention Test), EXIT 25 (Executive Interview 25), CCVLT (California Verbal Learning Test), CWT (Stroop Colour and Word Test), VLMT (Verbal Learning and Memory Test), CERAD (Consortium to Establish a Registry for Alzheimer's Disease), TAP (Test of Attentional Performance), TMPT (Trail-Marking Peg Test), HVLT-R (Hopkins Verbal Learning Test Revised), CNTABRTTT (Cambridge Neuropsychological Test Automated Battery Reaction-Time Touchscreen Task), VST (Victoria Stroop Test), SPMT (Scenery Picture Memory Test), DST (Digit Span Test), FAB (Frontal Assessment Battery at Bedside).

Regarding the main impact of the interventions, most studies (n = 34) showed a positive effect, at least in one CF domain. In terms of the results, considering the three categories of the cognitive domains presented above, it was verified that (1) the majority of the studies included variables that belonged to the EF and MF domains (n = 16) [30–32,34,38,40,42,45,47,48,51,52,56–58,63]. Six of these showed improvements in the outcomes related to both domains [30,31,34,38,40,52], seven showed improvements in one of the two domains [32,42,45,47,56,57,63], and only three did not report post-intervention improvements [48,51,58]. In one of these previous three studies [51], although they did not see significant improvements, the participants in the intervention group maintained the scores in almost all the main outcomes in a 2-year intervention period and the control group did not. (2) Ten studies analysed only EF domain outcomes [21,28,35,41,43,55,59–62], with a significant improvement in the one least assessed outcome. (3) Three only focused on GP [27,33,49] since they used an instrument that generated a final score, and just one did not report significant improvements [27]. (4) The remaining combined outcomes of EF and GP (n = 4) [36,39,50,53] or EF, MF, and GP (= 5) [29,37,44,46,54] were achieved. The interventions of Inoue, Kobayashi, Mori, Sakagawa, Xiao, Moritani, Sakane and Nagai [39], Liao, Chen, Hsu, Tseng, and Wang [46], and Kujawski, Kujawska, Kozakiewicz, Jakovljevic, Stankiewicz, Newton, Kędziora-Kornatowska, and Zalewski [44] showed improvements in all primary outcomes (Table 4).

## 4. Discussion

The main purpose of this study was to conduct a systematic review of observational and experimental studies examining the impact of PF programs on CF in elderly individuals. A total of thirty-eight articles involving 2389 participants (with 63% being female) were included in this review. Among them, thirty-four reported positive effects of physical training on at least one cognitive domain, including executive function, memory function, and general processing, following an intervention. Therefore, it is evident that older adults are able to enhance their CF domains through consistent participation in structured PF programs.

### 4.1. Cognitive Functions and Subdomains

We identified three primary categories of cognitive domains, with EF and MF being the predominant ones, followed by GP. Our findings align with the previous research, supporting the notion that physical exercise has a positive impact on the CFs of older adults [14,66]. Within the EF category, cardiorespiratory and muscle strength programs were observed to particularly benefit subdomains, such as working memory, attention, verbal fluency, reasoning, and processing speed. In the MF category, studies reported improvements in subdomains, including recognition, immediate recall, delayed recall, facial name recall, and paired associations. As for the GP category, enhancements were observed in immediate, short-term, and long-term memory parameters. It is worth noting that the studies included in this systematic review did not employ imaging tests to assess structural and functional changes in the brain following physical interventions. Nevertheless, our findings align with the review and meta-analysis studies [16,18], which support the connection between PF and neuroplasticity in old age.

We observed in the studies the combined use of specific instruments for the performances of EFs and MF, and we assumed this occurred because dysfunctions in EFs often preceded the decline in MF [67]. This can occur both in normal ageing and cases of preclinical dementia [68], consequently making it difficult to differentiate between the cognitive changes related to ageing and neurodegenerative diseases. EFs are recognised as higher cognitive processes linked to the prefrontal cortex of the brain; therefore, proper functioning favours goal-directed action [69], essential for self-control or self-regulation [70]. All of these are fundamental in older age for the planning and execution of instrumental activities of daily living [71,72]. Eleven studies reported a positive effect of the physical training program on MF, which suggested a relationship between PF and functional and

structural changes in the hippocampus. Episodic memory processes are subordinate to the hippocampus's anterior and posterior neocortical regions [73].

Nine studies indicated significant effects of physical training on GP. The finding corroborates recent reviews [74,75], attesting that PE-based training can create improvements to specific and global CFs. The screening identified three instruments to detect global cognitive changes: COGTEL, MMSE, and MoCA. All could provide aggregated information for cognitive performance. COGTEL had the advantage of being able to be applied in face-to-face and telephone interviews [76]. MoCA and MMSE were cognitive tests proposed for the early screening of cognitive impairments and dementia with an extensive use in older-adult investigations [77,78]. They are simple to understand, require little time to administer, and are easy to interpret.

*4.2. Intervention Program Types*

Cardiorespiratory training was prevalent in 19 studies. On a smaller scale, the use of walking on a treadmill and an ergometric bicycle was verified, while dancing was the most frequently used methodology. Of these, only Douka, Zilidou, Lilou, and Tsolaki [34] used a traditional style (traditional Greek dance), while the other studies adopted dance under the fitness methodology. A possible explanation for the greater use of dance aerobics was the ease this methodology offered to maintain and control the intensity of aerobic training. Comparatively, in traditional dance sections, the teaching methodology requires many pauses to explain and correct steps and gestures, often making it difficult to maintain a heart rate when shifting from moderate to vigorous intensities [79].

Review studies with meta-analyses pointed to dance as skilful training to promote cognitive improvements in older adults [80,81]. In addition to being attractive to this age group, rhythmic activities have the potential to associate with underlying mechanisms capable of inducing neural plasticity [79]. During a dance, there is an activation of the cardiac system, favouring the release of BDNF and insulin-like growth factor 1 (IGF-1), physiological mechanisms that are determinants for causing structural and functional alterations in the brain [6]. Moreover, a dance requires a constant adjustment of movements, often asynchronous between the legs, arms, head, and trunk in space following different rhythms [82]. Consequently, connectivity is strengthening that occurs between the two cerebral hemispheres [83]. Experimental studies on healthy older adults showed that dance training increased grey matter volume [52] and white matter [84]. It is worth mentioning that, although all nine included studies showed positive effects of dance in one of the three domains of cognition, we observed that in the majority, the intensity of the tasks needed to be more clearly detailed in the protocols.

The second most common PF program in this review was muscle strength. However, we observed a significant variability in the magnitude of procedures and results. Riegle van West, Stinear and Buck [57], Ladawan, Sungkamanee, Maharan, Amput, Srithawong, and Burtscher [45], and Macaulay, Pa, Kutch, Lane, Duncan, Yan, and Schroeder [47] observed improvement in EFs. The first two studies adopted meditative training (Poi, Tai Chi, and 4-movement Qigong), and the last used a machine-based exercise. We considered meditative training in the muscle strength category because of its relationship to strength development [85,86]. Improvements in CFs were verified in the study by Inoue, Kobayashi, Mori, Sakagawa, Xiao, Moritani, Sakane, and Nagai [39], who used a resistance program combining latex bands, squats, and Tai Chi. The other studies that used strength programs adopted different methodologies in two intervention groups to understand their effects on the cognitive domain. Gouveia É, Smailagic, Ihle, Marques, Gouveia, Cameirão, Sousa, Kliegel, and Siewiorek [37] verified the impacts of functional exercise and exergames fitness programs and training based only on functional exercise on short-term and long-term memory. However, the effects at the follow-up session (after four weeks) were observed only for the combined training group. Eckardt, Braun, and Kibele [35] also pointed out the variability in the outcomes. According to the authors, resistance training for instability improved working memory, processing speed, and response inhibition. On the

other hand, stable machine-based and stable machine-based adductor/abductor training did not generate improvements in EF. Kujawski, Kujawska, Kozakiewicz, Jakovljevic, Stankiewicz, Newton, Kędziora-Kornatowska, and Zalewski [44] analysed the influence of sitting callisthenic balance training versus resistance training, and the two programs positively influenced multiple cognitive domains. Finally, Castano, de Lima, Barbieri, de Lucena, Gaspari, Arai, Teixeira, Coelho, and Uchida [32] compared the effects of traditional resistance training versus resistance training combined with cognitive tasks, and the results showed only improvements in the resistance and cognitive training group.

Although these studies have shown positive effects of resistance and functional training for different domains of cognition, the results can be more consistent, suggesting a greater analysis of the protocols. The differences between the results were also observed in ten other studies [13,29,31,40,46,49,51,54,55,59] that adopted mixed methodologies (i.e., a combination of aerobic training, strength, flexibility, or balance).

Finally, four investigations used new technologies or exergames in their interventions. This methodology has been frequently used to improve the FPs and CFs of the older adult population [87]. However, the results of its real effectiveness for PFs are inconclusive due to the difficulty of controlling the intensity of the physical task [88]. On the other hand, this methodology has been shown to be superior to simple task training [89]. This training is also widely used in preventing falls, as it simultaneously requires several motor skills in accordance with cognition domains [87]. Our analysis observed in Ordnung, Hoff, Kaminski, Villringer, and Ragert [55], and Zhao, Zhao, Li, Zhao, Wang, Guo, Zhang, Sun, Ye, and Zhu [63] that the practice of exercise through exergames showed significant improvements in one or more EF domains. Liao, Chen, Hsu, Tseng, and Wang [46] found GP and EF improvements for older adults submitted to exergames and multicomponent training. However, only members of the exergame group improved their verbal and working memory at the end of the intervention. In Gouveia É et al.'s (2020) [37] study, exergames improved the patients' short- and long-term memory performances.

### 4.3. Length of Interventions, Frequency, and Time of Sessions

A high variability outcome was observed for the length of training programs (6 weeks–2 years), with three months prevailing. Only one study reported a two-year longitudinal design [51]. Regarding the training frequency, most activities occurred 2–3 times a week. Only one study was performed four times a week [42]. High heterogeneity was verified for the time of the training sessions, which varied from 30 to 90 min. When it came to guaranteeing the effect of physical exercise on CFs, it was fundamental that the training prescriptions considered aspects, such as frequency, intensity, type of exercise, session time, and length of intervention [90]. Frequency is an essential moderator for creating the stimuli necessary for neural plasticity [91]. A review study showed that comparatively moderate-frequency exercises (performed 3 to 4 times) had greater neural benefits than low-frequency exercises (performed 1 to 2 times) [92]. On the other hand, very intense training or overtraining could induce an increase in inflammatory cytokines and markers of oxidative stress, reducing the level of BDNF and impairing neural plasticity [93].

The possible explanations for the presented findings include the notion that regular exercise or training programs stimulate protective factors against age-related cognitive decline [16,94]. Exercise intensity plays a crucial role, as moderate to vigorous activities have demonstrated an increase in the levels of critical neurochemicals, such as the brain-derived neurotrophic factor (BDNF) and insulin-like growth factor-1 (IGF-1) [6,95]. Both factors promote synaptic plasticity and neuronal survival, which are vital for counteracting the decline in neural mass [96]. Our findings align with the previous research, including a previous meta-analysis that indicated structural changes in the hippocampal volume resulting from aerobic training [12]. Additionally, review studies and meta-analyses have highlighted the potential of resistance exercise training to induce structural and functional alterations in the brains of healthy, older adults [97,98].

### 4.4. Clinical Implications

The findings of this review demonstrate the potential for exercise-based interventions to mitigate cognitive decline in healthy, older adults, which is recognised as a significant measure of public health [99]. Notably, the effects of PF programs on the CF domains depend on each individual's biological, psychological, functional, and cognitive characteristics. Thus, issues, such as the length of the intervention, type of exercise, frequency, and duration of sessions, are fundamental [100,101]. Therefore, we suggest that future training protocols apply aerobic activities and resistance exercises with or without machines. Furthermore, we advise that training occurs 2–3 times a week (at a moderate PA level), in 60 min sessions. We also suggest that the interventions focus on developing strategies to promote the recruitment of men to ensure greater homogeneity in the analysis.

### 4.5. Strengths and Limitations

This study possesses notable strengths, particularly in its comprehensive coverage of current information from observational and experimental interventions utilising PF programs to potentially enhance the CFs of healthy, older individuals. Furthermore, our findings offer a valuable synthesis of information related to exercise frequency, exercise type, session duration, and intervention duration. However, there were certain limitations to our findings. Firstly, among the selected studies, there was an uneven distribution in terms of total participant numbers and gender and heterogeneity in the instruments employed for assessing cognitive performance. These disparities may have introduced biases to our results and conclusions. Secondly, we did not evaluate the intensity of the physical exercises applied, which was a crucial factor in stimulating neuroplasticity [16], due to the absence of this information in several included studies. Therefore, we recommend that future reviews focus on determining the optimal moderate- or high-intensity levels for promoting CF in the older, healthy population. Additionally, it is conceivable that the specific terms used to identify the relevant studies may have excluded certain articles, particularly those where the predefined terms were not present in the title and abstract. This potential limitation should be acknowledged.

## 5. Conclusions

The findings of this study consistently underscored the beneficial effects of involving older people in physical fitness interventions focusing on CF. Specifically, the cognitive domains expected to benefit more from a physical fitness program were EF, MF, and GP. Importantly for professionals who work with older people, it is crucial to remember that, among the various types of physical training employed, aerobic exercises were the most prevalent, emphasising activities, such as dancing, treadmill walking, and stationary cycling. Resistance training, using both machines and free weights and exergames, was also suggested as a common approach with an increased benefit. In order to keep an intervention based on the evidence, the revised studies typically spanned a duration of 3 to 6 months, with a frequency of two to three sessions per week, each lasting approximately 60 min. It is suggested that future studies may focus the analyses on the quality of the interventions, i.e., which new skill development/newer learning methods are the most beneficial for cognitive decline. The insights gleaned from this review can serve as valuable guidance for designing future interventions and contribute to formulating robust health policies aimed at promoting healthy ageing.

**Author Contributions:** M.A.T. and M.d.M.N. wrote the first draft of the manuscript. M.A.T. conducted the database searches. M.A.T., F.S. and S.M. conducted the data extraction. Methodological assessments were conducted by the same three authors. A.M., É.R.G. and A.I. wrote and reviewed the original article. All authors have read and agreed to the published version of the manuscript.

**Funding:** Swiss National Centre of Competence in Research LIVES—Overcoming vulnerability: life course perspectives, which was funded by the Swiss National Science Foundation (grant number: 51NF40-185901). Moreover, A.I. acknowledges the support received from the Swiss National Science

Foundation (grant number: 10001C_189407). We also acknowledge support from the Portuguese Recovery and Resilience Program (PRR), IAPMEI/ANI/FCT, under Agenda C645022399-00000057 (eGamesLab).

**Institutional Review Board Statement:** Not applicable.

**Informed Consent Statement:** Not applicable.

**Data Availability Statement:** No new data were created or analyzed in this study. Data sharing is not applicable to this article.

**Conflicts of Interest:** The authors declare no conflict of interest.

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
