# Peer review of "The Relationship between Physical Fitness and Cognitive Functions in Older People: A Systematic Review"

_sustainability, doi:10.3390/su152316314_

Round 1
Reviewer 1 Report
Comments and Suggestions for Authors
The authors of the manuscript “The relationship between physical fitness and cognitive functions among older people: a systematic review” has extensively collected data from research output as database and assessed the role of physical fitness on memory enhancement among older healthy individuals.
The study concludes saying it as inconclusive with few interesting data on role of dancing, aerobics, skill training in the betterment of cognition among aged individuals. However, the authors have not considered for new skill development/newer learning methods that could actually contribute for increase in hippocampal volume and gaining good synaptic plasticity. Few meta-analysis research has revealed the importance of new skill development/new language learning etc., as effective neuro-stimulant. In one of the recent meta-analysis, the importance of memory training or brain training in improving cognitive ability has been worked out. “Effectiveness of Cognitive Interventions in Older Adults: A Review. Eur J Investig Health Psychol Educ. 2020 Sep; 10(3): 876–898.”
The work is extraordinarily worked for evaluating the physical fitness to various cognitive intervention that could improve memory performance. However, there are few grammatical errors, duplication of words etc., that has to be corrected.
Also can make little precise write-up to make it easy for readers.
Scale of selecting the parameter for consideration as good, better and fair in the table given can be clarified.
Comments on the Quality of English Language
Language needs improvement for the current manuscript
Author Response
Cover letter
Dear Reviewer of Sustainability,
Following your previous suggestions, the authors have worked and improved the article entitled: The relationship between physical fitness and cognitive functions among older people: a systematic review. Below, we present the responses and changes made based on your suggestions. All changes have been underlined in yellow in the main manuscript. We believe that your comments contributed to strengthening the quality of the manuscript. We hope that you consider our paper for possible publication in the special issue entitled: Enhancing the Sustainability of Healthy Ageing and Long-Term Care through Innovations.
Suggestion 1: “…the authors have not considered for new skill development/newer learning methods that could actually contribute for increase in hippocampal volume and gaining good synaptic plasticity”.
R: We understand the rationale behind the reviewer's suggestion and appreciate the input that would be interesting to incorporate into a new study. However, the primary objective of this study was not to evaluate the quality of interventions in terms of effect size. Instead, the aim was to analyze the relationship between physical fitness and cognitive function. With this study, we aim to reinforce the message that there is a strong relationship between physical exercise and developing cognitive function in older age. While we observed a wide variability of program protocols, making it challenging to draw clear conclusions regarding newer learning methods, we found that interventions based on aerobic training, followed by resistance strength training and exergames, conducted over 3-6 months, ranging from 1-4 times a week with 30-90-minute sessions, seem to have a more significant impact on the following cognitive domains: executive function, memory function, and global cognition. We believe this study will guide professionals in planning future interventions to sustainably improve the older population's mental health. In the case of the suggested study, 'Effectiveness of Cognitive Interventions in Older Adults: A Review' (Sanjuán et al., 2020), they aimed to analyze the available evidence concerning the effect of cognitive interventions on improving or maintaining the general cognitive status of older adults who present different cognitive levels. No physical fitness intervention programs were analyzed.
Suggestion 2: “…there are few grammatical errors, duplication of words etc., that has to be corrected. Also can make little precise write-up to make it easy for readers.”
R: We have reviewed the text, and several corrections have been made to enhance the writing quality.
Suggestion 3: “Scale of selecting the parameter for consideration as good, better and fair in the table given can be clarified.”
R: The instrument for evaluating the quality of articles was now better described. Our strategy was to explain better which components of this instrument were used (p.3, lines 126 - 135). Also, in the results, we explain why each article is classified as weak, moderate, or strong in each category (p.4-5, lines 152 - 173).
Reviewer 2 Report
Comments and Suggestions for Authors
This manuscript is an example of a well performed review as methods are clearly stated and the synthesis of the findings is really well performed. This review sheds light in a topic that's it is still up for debate as a big part of society is yet to recognize the benefits of exercise in the elderly and are still leaning towards medication when exercise could have a greater and lasting benefit without secondary effects. In my humble opinion this review will help to lead researchers towards the necessary gaps of knowledge in this area needed.
Author Response
Cover letter
Dear Reviewer of Sustainability,
We are grateful to the reviewer for the positive evaluation of the paper. Based on suggestions from the other reviewers, we made slight changes to the article entitled: The relationship between physical fitness and cognitive functions among older people: a systematic review. All changes have been underlined in yellow. We believe that all the suggestions made contributed to strengthening the quality of the manuscript. We hope that you consider our paper for possible publication in the special issue entitled: Enhancing the Sustainability of Healthy Ageing and Long-Term Care through Innovations.
Reviewer 3 Report
Comments and Suggestions for Authors
The current systematic review investigated the impact of physical fitness programs on cognitive functions among healthy older adults. The results indicated that, indeed, physical fitness programs improve various domains of cognition in older adults.
This systematic review addresses a significant topic, which is relevant to the journal’s scope. It is well-conducted and it has the potential to provide positive feedback for future research.
The title is relevant and fits with the journal’s scope. The abstract is informative and follows the recommended structure for systematic reviews. It explicitly describes the objectives and the methodology used. The background knowledge is adequately described.
The introduction describes the rationale of the review and provides explicit statements of the objectives.
The methodology provides relevant information. However, it is essential to provide further information to meet the requirements that Prisma's statement suggests. Specifically, it would be important to better describe the decisions made about the screening and selection process. It is also essential to provide further information about the synthesis methods. The search strategy should be further analyzed. It is mentioned that “the keys search terms were linked by “AND”, and similar search terms were linked by “OR”. Can you provide more information about the search strings used? Did the authors use other boolean operators and how? It is mentioned that 1.418 studies were excluded at title and abstract level. It is important to better describe the processes of this stage of screening in the text. In addition, it is useful to better highlight the exclusion criteria.
Results are adequately described. Study selection, study characteristics, and risk of bias are analyzed. The table provides the essential information. I suggest providing a synthesis of the results before the discussion section.
The discussion includes information about the study’s strengths and limitations. The authors tried to describe how the various training programs improved different domains of cognition. This is an important point. They also compared these results with similar studies.
The conclusions section provides a summary of the main results. It would be helpful to provide additional critical interpretations of the results. Did the authors conclude which types of interventions, according to the selected studies, are the most beneficial for cognitive decline and why? In addition, it would be useful to highlight the importance of the current study's results for future research and mention more specific suggestions about the implications of the current study’s results.
Author Response
Cover letter
Dear Reviewers of Sustainability,
Following your previous suggestions, the authors have worked and improved the article entitled: The relationship between physical fitness and cognitive functions among older people: a systematic review. Below, we present the responses and changes made based on your suggestions. All changes have been underlined in yellow in the main manuscript. We believe that your comments contributed to strengthening the quality of the manuscript. We hope that you consider our paper for possible publication in the special issue entitled: Enhancing the Sustainability of Healthy Ageing and Long-Term Care through Innovations.
Suggestion 1: “…it is essential to provide further information to meet the requirements that Prisma's statement suggests. Specifically, it would be important to better describe the decisions made about the screening and selection process. It is also essential to provide further information about the synthesis methods.”
R: In order to meet the aforementioned suggestions, we change some points of the methodology: (1) We describe more explicitly the PICOS method used (p. 3, lines 97-100); (2) we added the exclusion criteria (p. 3, lines 100-102), (3) we reinforced, in the Screening Strategy and Study Selection, that duplicate articles were automatically removed by EndNote and were manually checked (p. 3, line 114); (4) at the point of data extraction and harmonization, we also made some small changes to the text (p.3, lines 121-122); (5) in figure 1, Flowchart of the study, we also changed the reasons why the articles were excluded, in order to make it more noticeable (See figure 1, p.4).
Suggestion 2: “The search strategy should be further analyzed. It is mentioned that “the keys search terms were linked by “AND”, and similar search terms were linked by “OR”. Can you provide more information about the search strings used?”
R: Indeed, this is relevant information; perhaps we could have presented it better. Therefore, we reformulated the second paragraph of the Seach Strategy, indicating in plain text which terms and Bolian operators were used (p.3, lines 102-106) and which strategy was used to define them (p.3, lines 106-107).
Suggestion 3: “Did the authors use other boolean operators and how?”
R: No, we don’t use other Boolean operators in the research. Due to the changes made to the suggestion above, we hope that this issue is more noticeable in the text.
Suggestion 4: “It is mentioned that 1.418 studies were excluded at title and abstract level. It is important to better describe the processes of this stage of screening in the text. In addition, it is useful to better highlight the exclusion criteria”.
R: Following these suggestions, we made some changes and corrections in the text. Regarding the description of the 1,418 studies excluded at the title and abstract level, we added the main reasons for this in the text (p. 3-4, lines 139-143). The exclude criteria we already highlighted in the response to suggestion 1.
Suggestion 5: “I suggest providing a synthesis of the results before the discussion section.”
R: We appreciate the reviewer's comment on this matter and agree with the importance of having a summary that contextualizes the study's main findings, making it easier for readers. However, the strategy adopted by the research team was to include this summary in the opening part of the discussion.
Suggestion 6: “It would be helpful to provide additional critical interpretations of the results”.
R: Additional justifications have been included in the discussion section (p. 17, lines 338- 345; p.19, lines 405-415).
Suggestion 7:” Did the authors conclude which types of interventions, according to the selected studies, are the most beneficial for cognitive decline and why?”
R: We understand the rationale behind the reviewer's suggestion and appreciate the input that would be interesting to incorporate into a new study. However, the primary objective of this study was not to evaluate the quality of interventions in terms of effect size. Instead, the aim was to analyze the relationship between physical fitness and cognitive function. With this study, we aim to reinforce the message that there is a strong relationship between physical exercise and developing cognitive function in older age. While we observed a wide variability of program protocols, making it challenging to draw clear conclusions regarding newer learning methods, we found that interventions based on aerobic training, followed by resistance strength training and exergames, conducted over 3-6 months, ranging from 1-4 times a week with 30-90-minute sessions, seem to have a more significant impact on the following cognitive domains: executive function, memory function, and global cognition. We believe this study will guide professionals in planning future interventions to sustainably improve the older population's mental health.
Suggestion 8: “…it would be useful to highlight the importance of the current study's results for future research and mention more specific suggestions about the implications of the current study’s results.”
R: The conclusion section was rewritten as follows to include the suggestions of the reviewer:
“The findings of this study consistently underscored the beneficial effects of involving older people in physical fitness interventions focusing on CF. Specifically, the cognitive domains expected to benefit more from a physical fitness program are EF, MF, and GP. Importantly for professionals who work with older people, it is crucial to remember that among the various types of physical training employed, aerobic exercises were the most prevalent, emphasizing activities such as dancing, treadmill walking, and stationary cycling. Resistance training, using both machines and free weights and exergames, was also suggested as a common approach with an increased benefit. In order to keep an intervention based on the evidence, the revised studies typically spanned a duration of 3 to 6 months, with a frequency of 2 to 3 sessions per week, each lasting approximately 60 minutes. It is believed that future studies may focus the analyses on the quality of the interventions, i.e., which new skill development/newer learning methods are the most beneficial for cognitive decline. The insights gleaned from this review can serve as valuable guidance for designing future interventions and contribute to formulating robust health policies aimed at promoting healthy ageing.” (p. 19-20, lines 446-460).